# Diagnosis of Depth of Submucosal Invasion in Colorectal Cancer with AI Using Deep Learning

**DOI:** 10.3390/cancers14215361

**Published:** 2022-10-31

**Authors:** Soichiro Minami, Kazuhiro Saso, Norikatsu Miyoshi, Shiki Fujino, Shinya Kato, Yuki Sekido, Tsuyoshi Hata, Takayuki Ogino, Hidekazu Takahashi, Mamoru Uemura, Hirofumi Yamamoto, Yuichiro Doki, Hidetoshi Eguchi

**Affiliations:** 1Department of Gastroenterological Surgery, Osaka University Graduate School of Medicine, Suita 565-0871, Japan; 2Department of Innovative Oncology Research and Regenerative Medicine, Osaka International Cancer Institute, Osaka 541-8567, Japan

**Keywords:** artificial intelligence, colorectal cancer, deep learning, endoscopy, submucosal invasion

## Abstract

**Simple Summary:**

In contrast to shallow submucosal invasion, colorectal cancer with deep submucosal invasion requires surgical colectomy. However, accurately diagnosing the depth of submucosal invasion via endoscopy is difficult. We developed a tool to diagnose the depth of submucosal invasion in early colorectal cancer using a convolutional neural network. The diagnostic accuracy of the constructed tool was as high as that of a skilled endoscopist. Endoscopic image recognition by deep learning might be able to predict the submucosal invasion depth in early-stage colorectal cancer in clinical practice.

**Abstract:**

The submucosal invasion depth predicts prognosis in early colorectal cancer. Although colorectal cancer with shallow submucosal invasion can be treated via endoscopic resection, colorectal cancer with deep submucosal invasion requires surgical colectomy. However, accurately diagnosing the depth of submucosal invasion via endoscopy is difficult. We developed a tool to diagnose the depth of submucosal invasion in early colorectal cancer using artificial intelligence. We reviewed data from 196 patients who had undergone a preoperative colonoscopy at the Osaka University Hospital and Osaka International Cancer Institute between 2011 and 2018 and were diagnosed pathologically as having shallow submucosal invasion or deep submucosal invasion colorectal cancer. A convolutional neural network for predicting invasion depth was constructed using 706 images from 91 patients between 2011 and 2015 as the training dataset. The diagnostic accuracy of the constructed convolutional neural network was evaluated using 394 images from 49 patients between 2016 and 2017 as the validation dataset. We also prospectively tested the tool from 56 patients in 2018 with suspected early-stage colorectal cancer. The sensitivity, specificity, accuracy, and area under the curve of the convolutional neural network for diagnosing deep submucosal invasion colorectal cancer were 87.2% (258/296), 35.7% (35/98), 74.4% (293/394), and 0.758, respectively. The positive predictive value was 84.4% (356/422) and the sensitivity was 75.7% (356/470) in the test set. The diagnostic accuracy of the constructed convolutional neural network seemed to be as high as that of a skilled endoscopist. Thus, endoscopic image recognition by deep learning may be able to predict the submucosal invasion depth in early-stage colorectal cancer in clinical practice.

## 1. Introduction

Colorectal cancer (CRC) has the highest cancer-related morbidity and the third highest mortality worldwide [1,2,3]. The prognosis of CRC varies based on the depth of tumor invasion. Accordingly, treatment is usually determined based on the degree of progression [3]. Almost all cases of early CRC are curable by surgery. Furthermore, endoscopic mucosal resection is possible without a colorectomy, depending on the depth of cancer invasion [4]. The Japanese guidelines divide submucosal (SM) CRC based on the depth of invasion into shallow SM invasion (SM-s; <1000 µm) and deep SM invasion (SM-d; ≥1000 μm). SM-s CRC without lymph node metastasis can be treated endoscopically, but SM-d CRC requires surgery with lymph node dissection [4]. Therefore, an incorrect assessment of the depth of invasion can result in the need to perform an additional colorectomy after initial endoscopic treatment. Such measures can lead to complications associated with endoscopic treatment, delayed surgical treatment, and increased cost [3].

Diagnosis of the depth of invasion of early CRC is difficult. The depth of invasion is assessed only by endoscopic findings in most cases. Although narrow-band imaging (NBI), staining, observation of surface and vascular structures with a magnifying endoscope, and the use of endoscopic ultrasonography have improved the accuracy of invasion depth diagnosis, [5,6,7,8,9] the accuracy reported by Japanese experts is only between 59% and 84% [10,11,12]. Hence, it is necessary to further improve the accuracy of diagnosis of the invasion depth of CRC using endoscopic findings.

Artificial intelligence (AI) has made dramatic progress in various fields in recent years, especially through machine learning (ML) [13,14,15,16]. Deep learning has also attracted substantial attention in the field of medicine [17,18,19]. To date, several AI diagnostic tools for CRC have been created by AI experts and other special facilities, such as companies and universities, in collaboration with doctors. However, we believe this approach is too inaccessible to contribute to diagnosis in general clinical practice. Therefore, in this study, we developed an AI auxiliary diagnostic support tool requiring only a personal computer and examined whether it could diagnose the depth of invasion as accurately as human experts.

## 2. Materials and Methods

### 2.1. Patients and Datasets

Patients pathologically diagnosed with SM-s or SM-d CRC after surgical resection (including additional resection after endoscopic resection) were retrospectively selected from 256 patients with early-stage CRC who underwent preoperative colonoscopy at the Osaka University Hospital and Osaka International Cancer Institute between January 2011 and December 2018. Endoscopic images from preoperative colonoscopy performed by a gastroenterologist or gastroenterological surgeon were selected from the electronic medical records at the Osaka University Hospital and Osaka International Cancer Institute, Osaka, Japan. The patient exclusion criteria were head invasion and insufficient pathology or preoperative laboratory data.

We collected the following information from the patients’ medical records: age, sex, levels of carcinoembryonic antigen and carbohydrate antigen 19-9, location of the primary tumor, and pathological findings. Clinicopathological factors were classified according to the 9th edition Union for International Cancer Control Tumor–Node–Metastasis classification.

### 2.2. Colonoscopy and Endoscopic Images

Colonoscopy was performed for screening or preoperative examination, and images were captured using standard endoscopes (CF-HQ290I, CF-H290I, PCF-H290I, CF-Q260AI, CF-H260AI, PCF-Q260AI; Olympus Medical Systems, Co., Ltd., Tokyo, Japan) and standard endoscopic video systems (EVIS LUCERA; Olympus Medical Systems). Images were included if they were taken with standard white light, chromoendoscopy using indigo carmine spraying, crystal violet, and NBI. Images were excluded if they were poor quality resulting from low insufflation of air, bleeding, halation, blur, defocus, or mucus. At least one CRC lesion was present in all the images, and multiple images were prepared for the same lesion to account for differences in angles, distance, and extension of the mucosa.

In addition, we prospectively tested the tool with images from 56 patients with suspected early-stage CRC who underwent preoperative colonoscopy at the Osaka University Hospital and Osaka International Cancer Institute between January and December 2018 as a test set and compared the accuracy with that of preoperative endoscopic diagnosis. The computer diagnosed the images of early CRC as either SM-s or SM-d. The obtained results were not reported to the endoscopists who performed the endoscopic examination to ensure that their diagnosis was not influenced. The colonoscopy image did not include personal information, and only the image of the tumor site was selected.

### 2.3. Analysis

Deep learning with AI was performed using MATLAB software (MathWorks, R2021a, Natick, MA, USA) by a convolutional neural network (CNN). During the deep learning process, we prevented the network from learning unnecessary information (e.g., backgrounds containing normal mucosa) by clipping the photographs containing the tumors in the form of squares [20]. We used occlusion which explicates the contribution of input images and evaluates an effect on classification results [21]. Differences in clinicopathological factors between the two groups were analyzed using the chi-square test or Fisher’s exact test. Continuous variables with parametric distributions were analyzed using Student’s t-test or analysis of variance. All statistical analyses were performed using JMP software version 16 (SAS Institute Inc., Cary, NC, USA).

## 3. Results

After selection, images from 196 of 256 patients were evaluated (Figure 1, Table 1). There were no significant differences between learning and validation sets (Appendix A). As the training image dataset for the construction of the CNN, 706 images were collected from 91 patients who underwent screening or preoperative examinations at the Osaka University Hospital between January 2011 and December 2015 and were indicated to have pathologically proven CRC (Figure 2, Table 2). To evaluate the diagnostic accuracy of the constructed CNN, an independent validation dataset of 394 images were collected from 49 patients who underwent screening or preoperative examinations at the Osaka University Hospital between January 2016 and December 2017.

Of the 394 images in the validation dataset, 98 images from 11 patients were pathologically proven as SM-s CRC, and 296 images from 38 patients were pathologically proven as SM-d CRC. The CNN identified 321 (87.2%) of the 394 validation set images as SM-d CRC: 63 images from tumors that were pathologically SM-s CRCs and 258 from tumors that were pathologically SM-d CRCs. In the retrospective validation set, the sensitivity of the CNN for diagnosing SM-d CRC compared to SM-s CRC was 87.2% (258/296), the specificity was 35.7% (35/98), the accuracy was 74.4% (293/394), and the area under the curve was 0.758 (Figure 3, Table 3). The percentage of images with incorrect AI diagnosis focused on other areas than the tumor area was 56.2% for SM-s CRC images and 60.5% for SM-d CRC images (Appendix A).

Re-verification was performed with the prospective test set of images taken from clinical practice. For the prospective test set, a total of 560 images were collected from 26 patients at the Osaka University Hospital and 30 patients at the Osaka International Cancer Institute with early CRC who underwent a preoperative colonoscopy. The sensitivity of the CNN was 75.7% (356/470) and positive predictive value was 84.4% (356/422) (Table 4A,B).

## 4. Discussion

Most endoscopy techniques use different lights and mucosal stains to diagnose the depth of invasion in early CRC. However, it is difficult to determine the depth of early CRC using endoscopy alone [7,22,23]. An inaccurate preoperative diagnosis can lead to the selection of inappropriate treatment methods, which may affect patient outcomes. If any tumor remains after mucosal resection and intestinal tract resection is not performed, the cancer may spread over time. Therefore, improving the diagnostic accuracy of early CRC invasion depth is necessary to avoid such complications.

The recent dramatic advancements in AI have improved its applicability in various fields of science [13,14]. ML is a method in which the computer is trained to find helpful similarities from given data. Deep learning is a type of ML that involves a learning structure of multiple layers to imitate the brain’s neural network, and it enables computers to analyze various training images and extract specific clinical features using a backpropagation algorithm. CNN is a type of deep learning network that enables the recognition of patterns through multilayer learning of image data and automatic extraction of image features [13,14]. Based on the accumulated clinical features, computers can be used to diagnose newly acquired clinical images using CNNs. Various types of neural networks have been developed, and CNNs are known to yield the best performance in the field of image recognition [16,17,18,19].

In this study, the accuracy of the CNN was 74.4%, equivalent to that of expert endoscopists. In previous studies, the accuracy of the diagnosis of depth of invasion in early CRC by endoscopists was 59–84% [10,22]. In our institute, the accuracy of preoperative diagnosis by endoscopists was 76%. In the validation set, the sensitivity and the positive predictive value of the CNN for early CRC were 87.2% and 74.4%, respectively. This shows that the CNN could diagnose the depth of invasion in early CRC as accurately as endoscopy experts. However, AI can only use the information in the image, not the image itself. Hence, it is better to obtain as much information as possible inside the image.

The details of the process of CNN deep learning and the part of the image the CNN reads and makes decisions about are not clear [24,25]. In this study, we constructed a program to visualize the part of the image being recognized and diagnosed by AI with a color map [20]. The areas that the AI focused on for diagnosis can be visualized with brighter color on the color map. We found that the AI’s image diagnosis took into account not only the tumor area but also information about the tumor surroundings. We prepared images that included both a close-up image of the tumor and the immediate tumor surroundings so that unnecessary information, such as the image background, would not affect the accuracy of the diagnosis. The created images were then learned by the CNN and a highly accurate diagnostic support tool was developed for endoscopic diagnosis of SM-d early CRC. When using colonoscopic images, close-up NBI images and staining methods such as indigo carmine and crystal violet are preferable for diagnosing the depth of invasion in early CRC compared to using only images captured with white light [7,22,26,27,28,29]. In this study, white light, NBI, and stained images were all used to assess SM invasion depth. Of the 1690 images in the total dataset, 1245 were white light images, 194 images were obtained from NBI, and 251 images were chromoendoscopy images. Although we tried to improve the accuracy of the deep learning tool using only NBI or stained images, the accuracy was too low. This can be attributed to the fact that there were few NBI and stained images for ML. All tumors had had white light images taken. Therefore, it was possible to obtain a sufficiently accurate diagnosis rate by repeating the learning process with the expanded dataset. Moreover, there was little difference in AI diagnosis when using white light and NBI or chromoendoscopy. Furthermore, we obtained nearly the same diagnostic accuracy as expert endoscopists using auxiliary tools for endoscopic image diagnosis. Hence, differences in diagnostic accuracy among endoscopic diagnostic doctors in clinical practice can be reduced using this tool. Further, it is expected that the accuracy of the CNN will increase as the number of learning cycles increase [15,24]. Therefore, future studies should increase the number of cases and allow the CNN to learn more endoscopic images. However, there are many reports on the usefulness of magnified images such as NBI and stained images compared to white light, and they are utilized in clinical practice. The depth of the lesion was diagnosed based on the macroscopic type of early CRC in this study. From the results of this study, we plan to collect uniform standards for endoscopic images (such as NBI, chromoendoscopy images) to examine diagnostic accuracy [30,31].

Several recent studies have reported the use of CNNs in medical diagnosis with high rates of diagnostic accuracy. However, as most studies are performed in specialized AI institutions with dedicated engineering systems, it takes a considerable amount of time to receive diagnostic results, which is not advantageous in clinical practice. In this study, we developed a cheap, accessible, and user-friendly tool for use in clinical practice to perform diagnosis using a CNN with a short turnaround time that requires only a personal computer. Both retrospective and test sets showed the high accuracy of our CNN approach. Furthermore, we obtained diagnostic accuracy comparable to that of experienced endoscopic diagnostic doctors using this auxiliary tool for endoscopic image diagnosis with the CNN. Hence, this CNN can be used to aid diagnosis in clinical practice.

This study has some limitations. One obvious limitation is the small number of SM-s cases in this study. There were twice as many SM-d cases as SM-s cases. As a result, the diagnostic accuracy of the CNN was lower in SM-s cases. On the other hand, it was a high positive predictive rate depending on the data set and a high false-positive rate. This study aimed that the endoscopic images were then analyzed by CNN and a helpful diagnostic tool was developed for endoscopic diagnosis of SM-d CRC. It should be used as supportive information to determine tumor depth preoperatively, because SM-s could be recognized as SM-d which is deeper CRC. Further, it is necessary to conduct an examination at multiple facilities in the future. Second, the color map visualization showed that not only the tumor area but also the tumor periphery was relevant for the AI diagnostic process. Therefore, it is necessary to examine whether the learning of not only endoscopic images of the tumor but also images including the tumor periphery affects the diagnostic accuracy. Finally, it was difficult to classify the gross appearance for colorectal cancer. We suspected that AI could be used to take some shortcut learning and diagnose depth of tumor invasion. However, this is the cohort study, and we collected data retrospectively. We carefully examined the study design regarding how to classify the tumor appearance of colorectal cancer. VR-Caps and CEP are software that can evaluate tissue volume using CT colonoscopy data [32,33]. Using these, we can obtain the best submucosal ground truth depth values and a large number of corresponding real images, and it is possible to construct new practical models. In this study, we evaluated the medical records as a retrospective analysis, and there are preoperative CT data, but not CT-colonography data. If we can handle the CT-colonography data, combining these applications with CT data and software would allow us to construct a practical model.

In conclusion, endoscopic image recognition with deep learning using AI may enable a more accurate diagnosis of SM invasion depth in early-stage CRC. It is necessary to examine the features of images that can improve the accuracy of diagnosis by collecting more cases.

## Figures and Tables

**Figure 1 cancers-14-05361-f001:**
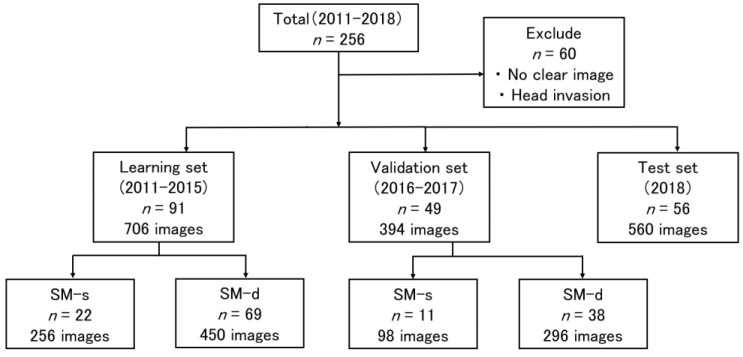
**Study population diagram.** Patients were classified into the learning set group, for which data was collected from 2011 to 2015, and the validation set group, for which data was collected from 2016 to 2017. SM-s: shallow SM invasion (<1000 μm), SM-d: deep SM invasion (≥1000 μm).

**Figure 2 cancers-14-05361-f002:**
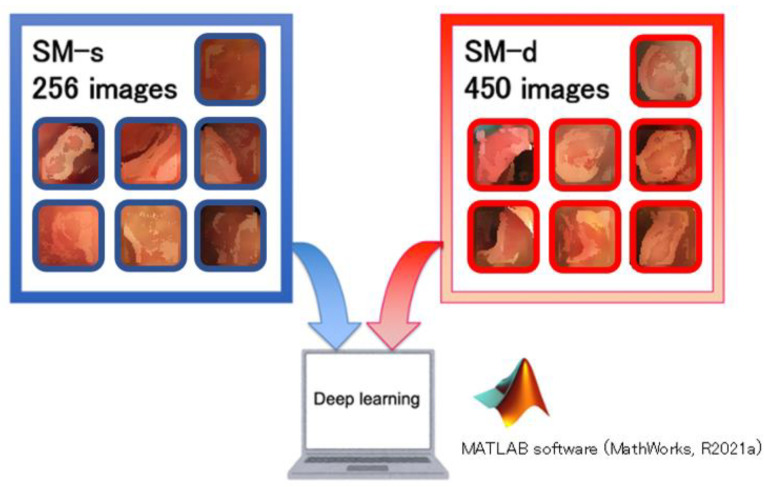
The computer used the 706 images learned by the computer for CNN analysis. The number of SM-s images was 256 and the number of SM-d images was 450.

**Figure 3 cancers-14-05361-f003:**
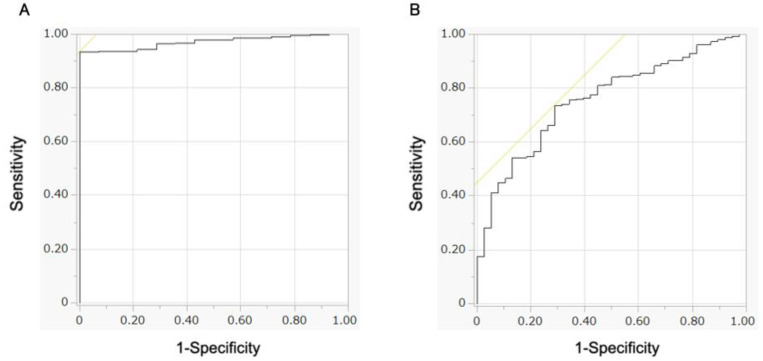
**Receiver operating characteristics curve.** (**A**) The area under the curve value of the learning set was 0.971. (**B**) The area under the curve value of the validation set was 0.758.

**Table 1 cancers-14-05361-t001:** Clinicopathological feature.

Variables	Learning Set (2011–2015)	Validation Set (2016–2017)
	*n* = 91	*n* = 49
Age, median (years) *	66 (41–86)	66 (35–84)
Sex (male/female)	59/32	28/21
Location (colon/rectum)	54/37	32/17
Greatest diameter, median (mm)	20 (0–50)	20 (6–66)
Preoperative CEA (≥5/5</NA) (ng/mL)	12/78/1	11/36/2
Preoperative CA19-9 (≥38/38</NA) (ng/mL)	5/85/1	3/44/2
Degree of differentiation (tub1/tub2/others)	57/30/4	29/19/1
Depth of invasion, median (µm) *	2000 (0–9000)	2200 (91–12,000)
Lymph node metastasis (+/−)	7/84	7/42
Lymphatic invasion (+/−)	24/67	16/33
Vascular invasion (+/−)	12/79	7/42
Budding grade (1/2, 3/NA)	54/16/21	37/8/4

* Continuous variables were evaluated. CEA, Carcinoembryonic antigen; CA19-9, Carbohydrate antigen 19-9; tub, Tubular adenocarcinoma; NA, Not applicable.

**Table 2 cancers-14-05361-t002:** Clinicopathological features (Learning set 2011–2015).

Variables	SM-s (*n* = 22)	SM-d (*n* = 69)	*p*-Value
Age, median (years) *	66 (41–86)	66 (35–84)	0.846
Sex (male/female)	15/7	44/25	0.706
Location (C/A/T/D/S/R) ^a^	1/2/1/0/7/11	7/14/7/5/10/26	NA
Degree of differentiation (tub1, 2/others)	22/0	65/4	0.248
Lymphatic invasion (+/−)	4/18	20/49	0.317
Vascular invasion (+/−)	2/20	10/59	0.514
Budding grade (1/2, 3) **	13/2	41/14	0.322

* Continuous variables were evaluated. ** Not applicable: 7 in SM-s, 14 in SM-d. ^a^ Tumor located at cecum (C), ascending (A), transverse (T), descending (D), sigmoid (S), colon or rectum (R) was determined according to the Japanese Classification of Colorectal Carcinoma (9th ed). NA: Not assessed; tub, Tubular adenocarcinoma; SM-d, deep submucosal invasion; SM-s, shallow submucosal invasion.

**Table 3 cancers-14-05361-t003:** Results of CNN learning from the validation set.

		AI Diagnosis	
		SM-s	SM-d
**Pathological classification**	SM-s *n* = 11 98 images	35 images	63 images
	SM-d *n* = 38 296 images	38 images	258 images

Sensitivity: 87.2% (258/296), positive predictive value: 80.4% (258/321), accuracy: 74.4% (293/394). CNN, convolutional neural network; SM-d, deep submucosal invasion; SM-s, shallow submucosal invasion.

**Table 4 cancers-14-05361-t004:** Prospective test set. (A) Results of CNN learning. (B) Diagnostic accuracy: Clinical diagnosis vs. AI diagnosis.

**(A)**					
		*n* = 56 560 images			
**Age, median (years) ***		63.8 (38–82)			
**Sex (male/female)**		30/26			
**Location (C/A/T/D/S/R) ^a^**		5/7/9/5/8/22			
**Degree of differentiation (tub1/tub2/others)**		37/17/2			
**Lymphatic invasion (+/−)**		11/45			
**Vascular invasion (+/−)**		5/51			
**Budding grade (1/2, 3/NA)**		40/14/2			
**(B)**					
		**Clinical diagnosis**		**AI diagnosis**	
		**SM-s**	**SM-d**	**SM-s**	**SM-d**
**Pathological classification**	SM-s *n* = 9 90 images	*n* = 5	*n* = 4	24 images	66 images
	SM-d *n* = 47 470 images	*n* = 17	*n* = 30	114 images	356 images

* Continuous variables were evaluated. ^a^ Tumor located at cecum (C), ascending (A), transverse (T), descending (D), sigmoid (S), colon or rectum (R) was determined according to the Japanese Classification of Colorectal Carcinoma (9th ed). Clinical diagnostic sensitivity: 75.7% (356/470), positive predictive value: 84.4% (356/422). AI diagnostic sensitivity: 63.8% (30/47), positive predictive value: 88.2% (30/34). AI, artificial intelligence; SM-d, deep submucosal invasion; SM-s, shallow submucosal invasion. AI, artificial intelligence; CNN, convolutional neural network; tub, Tubular adenocarcinoma; NA, not assessed.

## Data Availability

The datasets used and/or analyzed during the current study are available from the corresponding author upon reasonable request.

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
