# Peer review of "Diagnosis of Depth of Submucosal Invasion in Colorectal Cancer with AI Using Deep Learning"

_cancers, 2022, doi:10.3390/cancers14215361_

Round 1
Reviewer 1 Report
Authors used CNN based AI auxiliary diagnostic support tool to diagnose the depth of submucosal invasion in CRC. Researchers used the retrospective analysis where they used images from confirmed CRC patients as well as prospectively tested the AI model in suspected CRC patients.
Minor comment:
It is well known that AI model are only as good as the training data it was provided, therefore performance and generalizability of any model is a major challenge in application of these model in diagnosis. In many cases, the broad adaptation of AI models is hindered by the nature of the training datasets it is trained upon.
In the abstract section, author concluded by stating “The diagnostic accuracy of the constructed convolutional neural network was as high as that of a skilled endoscopist.” Such strong conclusion must be made carefully with robust analysis because this could be case by case basis. The AI tool discussed here may not have same level of specificity and sensitivity when applied in different imaging data source from other demographics or different institutions.
Major comments:
1. Author used datasets from 2011-2015 as learning set and 2016-2017 for validation set, yet mentioned the samples were selected randomly. This does not seem like random sampling. To test robust analysis, it is recommended to perform cross validation or similar approach.
2. Discussion section of manuscript id not properly address the limitations of the tool such as false-positive and false-negatives as well as did not provide the caution of using AI tool or its generalizability.
3. The tool was trained in small sample size and data from single institution or demographic. It is very premature to claim the robustness of the tool without testing the tool in multi-institutional samples with wider demographic or more heterogeneous sample.
4. There are many other models than CNN models. Author should also test the data in other ML models.
I highly suggest author to address major comments by:
1. Additional testing of the tool in more heterogenous data, more sample, and proper sample selection techniques.
2. Significantly increase sample size for training and test datasets while including broader demographics.
3. Test the superiority of CNN model over other model in their datasets.
4. Revising the strong statements with more practical statements.
Author Response
Reviewer 1
Thank you very much for your careful review of our old manuscript. According to the reviewer’s comments, we revised our manuscript as marked in red as follows.
Minor comment:
In the abstract section, statement
Thank you for your helpful comments and suggestions. As reviewer#1 pointed out, the statement should be rephrased, not strong as follows. According to the reviewer’s comments, we revised our manuscript as marked in red.
Page 2, lines 18-19.
“The diagnostic accuracy of the constructed convolutional neural network seemed to be as high as that of a skilled endoscopist.”
Major comments:
(1) Datasets selection (not randomly)
Thank you for your careful review and comments. Learning and Validation set included all target cases in the period. Test set was prospectively collected, and we re-analyzed all 56 target cases in 2018 without random selection. According to the reviewer’s comments, we revised our manuscript as marked in red.
Page 4, lines 19-22.
“In addition, we prospectively tested the tool with images from 56 patients with suspected early-stage CRC who underwent preoperative colonoscopy at the Osaka University Hospital and Osaka International Cancer Institute between January and December 2018 as a test set and compared the accuracy with that of preoperative endoscopic diagnosis.”
Page 5, lines 25-28.
“For the prospective test set, total 560 images were collected from 26 patients at the Osaka University Hospital and 30 patients at the Osaka International Cancer Institute with early CRC who underwent preoperative colonoscopy. The sensitivity of the CNN was 75.7% (356/470) and positive predictive value was 84.4% (356/422)”
Table 4. Prospective test set
- B) Diagnostic accuracy: Clinical diagnosis vs AI diagnosis.
B.
|
|
Clinical diagnosis |
AI diagnosis |
|||
|
SM-s |
SM-d |
SM-s |
SM-d |
||
|
Pathological classification |
SM-s n=9 90 images |
n=5 |
n=4 |
24 images |
66 images |
|
SM-d n=47 470 images |
n=17 |
n=30 |
114 images |
356 images |
|
Clinical diagnostic sensitivity: 75.7% (356/470), positive predictive value: 84.4% (356/422)
AI diagnostic sensitivity: 63.8% (30/47), positive predictive value: 88.2% (30/34)
(2) Discussion section: Limitations of the tool such as false-positive and false-negatives and the caution of using AI tool or its generalizability.
Thank you for your important comments. This study aimed that collected images were then analyzed by CNN, and a helpful, accurate diagnostic support tool was developed for endoscopic diagnosis of SM-d CRC. As reviewer indicated, our results of the analysis showed a high positive-predictive rate depending on the data set and a high false-positive rate. It should be used as “information” to determine tumor depth preoperatively “carefully,” because SM-s could be recognized as SM-d which is deeper CRC. According to the reviewer’s comments, we revised our manuscript as marked in red.
Page 7, lines 30-34.
“On the other hand, it was a high positive predictive rate depending on the data set and a high false-positive rate.
This study aimed that the endoscopic images were then analyzed by CNN and a helpful diagnostic tool was developed for endoscopic diagnosis of SM-d CRC. It should be used as supportive information to determine tumor depth preoperatively, because SM-s could be recognized as SM-d which is deeper CRC.”
(3) Trained in small sample size and data from single institution.
Thank you for your helpful comments and suggestions. We collected additional data. We added another institute. In the revision, we examined two facilities, and still also think it is necessary to conduct an examination at multiple facilities in the future. According to the reviewer’s comments, we revised our manuscript as marked in red.
Page 7, lines 34-35.
“And it is necessary to conduct an examination at multiple facilities in the future.”
(4) Test the data in other ML models.
Thank you for your helpful suggestions. In this study, we used alexnet with Matlab platform. And we performed a machine learning for the diagnosis using endoscopic images. The graphical data (the brightness) of the endoscopic images (measured by the ImageJ software; https://imagej.nih.gov/ij/index.html). Cutoff was determined from ROC curves, and whether SM-s or SM-d could be diagnosed was examined. As a result, the positive-predictive value was 76.1% and sensitivity was 25.1%, that were lower than our AI diagnosis.
Supplementary Table (not mentioned in the paper)
|
|
Image J diagnosis |
||
|
SM-s |
SM-d |
||
|
Pathological classification |
SM-s n=9 90 images |
53 images |
37 images |
|
SM-d n=47 470 images |
352 images |
118 images |
|
Cutoff: 139
Sensitivity: 25.1% (118/470), positive predictive value: 76.1% (118/155)

Reviewer 2 Report
Diagnosis of depth of submucosal invasion in colorectal cancer with AI using deep learning
In this manuscript, authors reported the application of deep learning in the early detection of submucosal invasion in colorectal cancer. Usually, several methods in addition to endoscopic examination were previously used to diagnose and examine shallow or depth invasion of colorectal cancer. However, these methods are tedious and not 100% accurate. The current article describes well classifying the submucosal invasion i.e., shallow or depth using AI-deep learning approach. Based on current study, endoscopic alone examination was enough to classify tumor status more accurately and specifically. Although, these samples are not validated against enough sample numbers for both classifiers, this article shows the feasible approach to mark the early detection of tumor progression.
However, manuscript lacks minor controls and analysis that must be addressed to support their claims made in the paper prior to publication.
1. Authors have taken <1000 images to conclude their data. Please mention about other cases where these number of samples were used for training or test sets. Less learning might also give false positive to the validation set. How authors think about it?
2. Lymph node examination is a standard approach to distinguish between shallow or deep tumor progression/invasion. Based on deep learning approach, author did not comment about what basis (what label or features) program classify tumors progression: deep or shallow.
Authors has rarely mentioned or compared their concepts with other already available detection methods. It would be great to mention how your data processing is better or more feasible than others.
Author Response
Thank you very much for your careful review of our old manuscript. According to the reviewer’s comments, we revised our manuscript as marked in red as follows.
(1) More images (additional data and reanalysis)
Thank you for your helpful comments and query. We analyzed a total of 1690 images used in the Learning and Validation set. We have added data from other institutions, including cases from January 2011 to December 2018. The results of the analysis showed a high positive-predictive rate depending on the data set and a high false-positive rate.
It should be used as supportive information to preoperatively determine tumor depth carefully, because SM-s could be identified as SM-d (deeper) by AI.
According to the reviewer’s comments, we revised our manuscript as marked in red.
Page 7, lines 30-34.
“On the other hand, it was a high positive predictive rate depending on the data set and a high false-positive rate.
This study aimed that the endoscopic images were then analyzed by CNN and a helpful diagnostic tool was developed for endoscopic diagnosis of SM-d CRC. It should be used as supportive information to determine tumor depth preoperatively, because SM-s could be recognized as SM-d which is deeper CRC.”
(2) Comment about basis, program classify tumors progression: deep or shallow, and available detection methods
Thank you for your careful review, important comments and suggestions. In previous reports, NBI and chromoendoscopy have a high accuracy about 90% and a positive-predictive value about 80% in diagnosing the depth of invasion. Our results showed a precision of 67.9% and a positive-predictive value (84.4%). In endoscopic diagnosis, the macroscopic type of the tumor is one of the factors to diagnose the depth determining early colorectal cancer. In this study, the depth of the lesion (deep or shallow) was diagnosed based on the macroscopic type by AI objectively. We revised our manuscript as marked in red.
Page 7, lines 16-17.
“The depth of the lesion was diagnosed based on the macroscopic type of early CRC in this study.”

Round 2
Reviewer 1 Report
In the revised version, the authors have tried to address the major comments.